# In Vitro Digestion Assessment (Standard vs. Older Adult Model) on Antioxidant Properties and Mineral Bioaccessibility of Fermented Dried Lentils and Quinoa

**DOI:** 10.3390/molecules28217298

**Published:** 2023-10-27

**Authors:** Janaina Sánchez-García, Sara Muñoz-Pina, Jorge García-Hernández, Amparo Tárrega, Ana Heredia, Ana Andrés

**Affiliations:** 1Instituto Universitario de Ingeniería de Alimentos (FoodUPV), Universitat Politècnica de València, Camino de Vera s/n, 46022 Valencia, Spain; jasanga7@doctor.upv.es (J.S.-G.); anhegu@tal.upv.es (A.H.); aandres@tal.upv.es (A.A.); 2Centro Avanzado de Microbiología de Alimentos (CAMA), Universitat Politècnica de València, Camino de Vera s/n, 46022 Valencia, Spain; 3Instituto de Agroquímica y Tecnología de Alimentos (IATA-CSIC), Agustín Escardino 7, 46980 Valencia, Spain; atarrega@iata.csic.es

**Keywords:** *Pleurotus ostreatus*, phenolic profile, antioxidant activity, total phenol content, phytic acid

## Abstract

The growing number of older adults necessitates tailored food options that accommodate the specific diseases and nutritional deficiencies linked with ageing. This study aims to investigate the influence of age-related digestive conditions in vitro on the phenolic profile, antioxidant activity, and bioaccessibility of minerals (Ca, Fe, and Mg) in two types of unfermented, fermented, and fermented dried quinoa and lentils. Solid-state fermentation, combined with drying at 70 °C, significantly boosted the total phenolic content in Castellana and Pardina lentils from 5.05 and 6.6 to 10.5 and 7.5 mg gallic acid/g dry weight, respectively, in the bioaccessible fraction following the standard digestion model, compared to the unfermented samples. The phenolic profile post-digestion revealed elevated levels of vanillic and caffeic acids in Castellana lentils, and vanillic acid in Pardina lentils, while caffeic acids in Castellana lentils were not detected in the bioaccessible fraction. The highest antioxidant potency composite index was observed in digested fermented dried Castellana lentils, with white quinoa samples exhibiting potency above 80%. Mineral bioaccessibility was greater in fermented and fermented dried samples compared to unfermented ones. Finally, the digestive changes that occur with ageing did not significantly affect mineral bioaccessibility, but compromised the phenolic profile and antioxidant activity.

## 1. Introduction

The ageing population is predominantly due to decreased fertility rates and increased life expectancy. It is projected that by 2050, the number of individuals over the age of 60 will reach approximately 2 billion, representing 22% of the global population, with the majority residing in developing nations [1]. Therefore, the forthcoming expansion of older adults’ population will cause substantial increased demands for food products that are specifically formulated to meet their preferences and nutritional requirements. Ageing frequently causes digestive disorders related to changes in the oral cavity, including tooth loss and wearing dentures, gingivitis, and reduced saliva production. Furthermore, reduced sense of taste and smell can decrease food palatability and increase inappetence, leading to changes in the type and quantity of food consumed [2]. Gastric emptying slows down, and the gastric lipase and pepsin enzyme secretions are reduced, leading to an alkalisation of the gastric environment. Furthermore, peristalsis in the small intestine decreases, resulting in reduced secretion of pancreatic enzymes and bile salts [3,4,5]. These gastrointestinal tract alterations may contribute to age-related malnutrition, causing deficiencies in micronutrients, particularly minerals, which can lead to functional decline, fragility, and difficulty in maintaining independent living [6]. Furthermore, phytochemicals such as polyphenols may be considerable in chronic diseases, including cardiovascular disease, type II diabetes, cancer, osteoporosis, and neurodegenerative diseases [4]. Therefore, it is crucial to include antioxidants and minerals in one’s diet to maintain brain function, support bone and teeth health, aid cellular and thyroid metabolism, and strengthen the immune system of older individuals [7,8,9,10].

In this scenario, assessing the impact of common gastrointestinal conditions in older adults on the digestibility of novel ingredients with enhanced antioxidant properties and improved digestibility is important. This assessment is crucial for creating new highly nutritious foods adapted to older adults. Therefore, grains and seeds, such as lentils or quinoa, could be considered good candidates as raw materials for developing protein-rich functional ingredients. The antioxidant activity of these plant materials is associated with a high content of phenolic compounds. Lentils have a higher reported total phenolic content (7.53 mg GAE/g sample) than other legumes, including peas, chickpeas, soybeans, red kidney, and black beans [11]. Quinoa has a total phenolic content (TPC) of 5.18 mg GAE/g sample [12], possessing antioxidant properties that are more effective than those of other cereals and pseudocereals, such as brown rice, millet, whole wheat, barley, oats, rye, Job’s tears, corn, and amaranth [13]. Furthermore, some of the phenolic compounds present in lentils are flavonoids, including kaempferol glycosides, catechin/epicatechin glycosides, and procyanidins [14]. Phenolic acids, namely vanillic acid, ferulic acid, and their derivatives, and flavonoid compounds such as quercetin, kaempferol, and their glycosides, were found in quinoa [12,15]. Nevertheless, consumption of these compounds may not offer full health advantages because of factors such as antinutrients and limited digestibility. Solid-state fermentation (SSF), however, enhances the antioxidant properties and nutritional quality of diverse legumes and cereals. Thus, it is possible for the TPC of fermented plant materials to increase because of the release of phenolic compounds. These compounds are produced due to the structural breakdown of the cell wall after fungal colonisation, the action of ligninolytic and hydrolytic enzymes, or the synthesis of soluble phenolic compounds conducted by the fermentative micro-organism [16]. It is important to note that this increase is not subjective, but based on scientific evidence and observations. The variability in antioxidants altered by SSF relies on the binomial substrate–microorganism and process variables, precluding the generalisation of findings across studies. Furthermore, studies showed that the TPC of fermented black bean, kidney bean, and oat samples increased up to twice as much compared to unfermented digested samples after gastrointestinal digestion, mimicking the healthy adult digestion model [17].

The bioavailability of minerals in plant materials is relatively low due to certain molecules, such as phytates or phenols, forming complexes [18]. However, SSF was discovered to decrease phytates by endogenous phytase action, which is activated during fermentation. This leads to mineral release and increased bioavailability [19].

This study aims to analyse the effect of common age-related digestive conditions on the phenolic profile and antioxidant activity of the bioaccessible fraction together with the bioaccessibility of minerals (Ca, Fe, and Mg) of unfermented, fermented, and fermented dried (hot air drying or lyophilisation) quinoa (white and black) and lentils (Castellana and Pardina). Furthermore, all samples were subjected to in vitro digestion under healthy standard GI conditions for comparison.

## 2. Results and Discussion

### 2.1. Impact of GI Conditions on the Release of Phenols and Antioxidant Activity of Unfermented Fermented, and Fermented Dried Lentils and Quinoa

TPC and antioxidant activity changes during digestion were analysed and shown in Figure 1. In undigested samples, SSF and hot air drying at 70 °C resulted in an increase in TPC content in quinoa and lentils, compared to unfermented flours. The reasons behind this increase are elaborated in detail by Sánchez-García et al. [20,21]. During digestion, the gastrointestinal process induced a rise of free phenols in the bioaccessible fraction, regardless of the simulated conditions (standard or older adult) and the processing conducted to obtain flour from the substrates. However, this increase was more significant in lentils, as compared to quinoa, and in the fermented samples (FPL, FCL), compared to unfermented ones (UFPL, UFCL). Furthermore, samples fermented and dried at 70 °C (FPL-70, FCL-70) showed the highest TPC in the bioaccessible fraction. The optimal conditions for extracting phenols were pH, enzymatic activity, temperature, and stirring during the digestion process.

When TPC in the bioaccessible fractions was measured using two digestive models, the TPC in the fermented dried samples remained largely unaltered. Due to the common altered gastrointestinal conditions that appear with ageing, fermented dried lentils at 70 °C are especially interesting in terms of their bioaccessible TPC.

Tungmunnithum et al. [22] reported similar results in 10 bean varieties consumed in Thailand, with an increase in phenolic and flavonoid content associated with digestion. Phenol and flavonoid content rose with digestion but decreased during bean cooking. After gastrointestinal digestion, the TPC and total flavonoid content increased between 9% and 190%, and 4% and 266%, respectively, across different varieties. Physiological factors, such as digestive enzymes, bile salts, and pH, play a crucial role in the release of these compounds. Certain phenolic compounds are not present in free form in grains or seeds but are bound to the cell wall, creating macromolecular complexes. In addition to gastric digestion, a low pH causes an increase in polyphenols in their undissociated form, facilitating their release from the food matrix into the aqueous phase [23,24]. However, the links between phenolic compounds and carbohydrates are reduced during intestinal digestion by the action of pancreatic enzymes, bile salts, and a neutral pH (6.9) [23,25].

The phenolic fraction’s chromatographic analysis presented distinct profiles among the bioaccessible fractions based on both flour variety and the digestive model, as shown in Table 1, Table 2, Table 3 and Table 4. Previously published studies [20,21] also performed the same chromatographic analysis on all samples before digestion. Furthermore, the chromatograms of the phenolic profile corresponding to unfermented, fermented, and fermented dried Pardina lentil samples after gastrointestinal digestion, both under the healthy adult (standard) and the older adult digestion model, can be found in the Appendix A. Upon comparison of substrates after in vitro digestion, the bioaccessible fractions of Castellana lentils demonstrated a greater abundance of vanillic and caffeic acids, whereas Pardina exhibited a great abundance of 4-O-caffeoylquinic and vanillic acids. White and black quinoa boasted higher amounts of gallic and vanillic acids, as well as quercitrin, compared to lentils. However, the quantities of these compounds differed depending on the treatment undergone by the flours. The bioaccessible portion of the unfermented flours contained lower levels of these compounds compared to their fermented counterparts, particularly those exposed to hot air drying. Vanillic and caffeic acids demonstrated a greater increase in the Castellana lentil, rising from 6.2 to 20 µg/g dry basis and from 3.4 to 10.8 µg/g dry basis, respectively. In contrast, vanillic acid increased from 7.6 to 20.7 µg/g dry basis in Pardina lentils. Furthermore, an increase in gallic acid was observed in both white and black quinoa samples fermented and dried at 70 °C. The increase in gallic acid was apparent and rose from 20 to 139 µg/g dry basis in white quinoa and from 30 to 42 µg/g dry basis in black quinoa. Consequently, SSF plus drying facilitates the liberation of specific phenolic acids and flavonoids, resulting in their incorporation into the water-soluble bioaccessible fraction during gastrointestinal digestion.

Older adult simulated conditions resulted in a significant reduction in the variety and number of phenolic compounds present in the bioaccessible fraction when compared to standard digestive conditions. However, chromatographic analysis did not detect all compounds in samples digested under the older adult model. In contrast, the same compounds were found in the bioaccessible fraction obtained using the standard model. This applies to vanillic and 4-hydroxybenzoic acids in the Castellana lentil. Phenolic compounds, found in various plant sources, can have preventive health benefits for humans. The extent of these benefits depends on the compounds’ structure, such as their degree of glycosylation or acylation, molecular size, solubility, and conjugation with other phenols. These factors ultimately determine their absorption and metabolism [26]. Vanillic acid [27,28], caffeic acid [29,30], and gallic acid [31,32] are widely recognised as common phenolic acids that exhibit diverse chemical and pharmacological properties, such as analgesic, anticancer, anti-inflammatory, antioxidant, antimicrobial, cardioprotective, and neuroprotective activities.

The antioxidant activity of the bioaccessible fraction was assessed through three assays: ABTS, FRAP, and DPPH. Table 5 displays the indexes for each assay and the antioxidant potency composite index (APCI). The ABTS-antioxidant activity increased following in vitro digestion, whereas no changes or slight decreases were observed in the FRAP and DPPH assays. Significant differences were found in the antioxidant activities of the bioaccessible fraction, with lower values when older adult conditions were used. The ABTS and DPPH antioxidant capabilities were reduced between 1% and 50%. However, the FRAP assay showed a decrease only in lentils, whereas quinoa showed an increase ranging from 10% to 70% for the digesta values of the older adult model compared to those of the standard model. Gallego et al. [33] discovered similar results when evaluating the effect of cooking different legume pastes on antioxidant activity after gastrointestinal digestion using the DPPH, ABTS, and FRAP methods. The study indicated a noteworthy improvement in the antioxidant activity of lentil pastes, reaching 12-fold greater levels than their original undigested content. However, they also found up to a four-fold reduction in pea paste using the DPPH method. The authors explained that these differences were due to the activity of enzymes in the gastrointestinal system. These enzymes promote the breakdown of proteins and peptides, resulting in the release of amino acids and phenolic compounds, and the exposure of internal groups. These factors impact the amount, dimensions, and physicochemical features of these compounds and influence the antioxidant potential. Koehnlein et al. [34] suggested that the high antioxidant capacity of cereals and legumes following gastrointestinal digestion may be due to the partial hydrolysis of total phenols and an increase in their content. Furthermore, the hydroxyl groups on the aromatic rings of the phenolic compounds may be deprotonated. Of all the treatments, flours that were fermented and dried 70 °C displayed the greatest antioxidant activity after gastrointestinal digestion. The fermented flours derived from Castellana lentil (FCL-70) and white quinoa (FWQ-70) demonstrated greater antioxidant capacity with an APCI exceeding 90% and 80%, respectively. Consequently, the results confirm the effectiveness of SSF followed by hot air drying (70 °C) in generating flours that boast an improved functionality of the bioaccessible fraction.

The effect of gastrointestinal conditions shows different trends depending on the pre-treatment of the food and the substrate itself, as well as the methodology used to measure the antioxidant activity. A significant reduction is only observed for DPPH in Pardina lentils and for ABTS in black quinoa after using the older adult model. However, white quinoa and Castellana lentils maintain or even increase the values reported by the control model. These data could help develop new products for this population group, because a high antioxidant capacity is necessary for good health.

### 2.2. Impact of GI Conditions on the Bioaccessibility of Phytic Acid and Minerals of Unfermented, Fermented, and Fermented Dried Lentils and Quinoa

Minerals are inorganic substances found in all tissues and bodily fluids, which are vital for maintaining specific physicochemical processes essential for life [35,36]. They have structural functions involving the skeleton and soft tissues and regulatory functions including neuromuscular transmission, blood clotting, oxygen transport, and enzyme activity [37]. Legumes and pseudocereals are excellent sources of minerals such as calcium, iron, zinc, potassium, and magnesium [38,39]. The mineral content of unfermented, fermented, and fermented dried samples (Mg, Ca, and Fe) was evaluated pre- and post-gastrointestinal digestion, as shown in Table 6. Undigested samples revealed that SSF and ensuing drying caused an increase in Mg and Ca contents, with increases in Mg ranging from 1% to 20% and Ca from 12% to 59%. Significant differences in Ca content were found in all samples, whereas significant differences in Mg content were found only in Castellana lentil and white quinoa samples dried at 70 °C and lyophilised. In contrast, SSF decreased the Fe content in all samples between 2% and 11%, with a significant difference in Pardina lentil and white quinoa samples. Furthermore, the drying process increased the Fe content between 2% and 23% only in white and black quinoa samples, with significance in the fermented samples dried at 70 °C.

When comparing the undigested and digested samples, the findings indicate that the digestive process led to a decrease in mineral content (Mg, Ca, and Fe) in the bioaccessible fraction. Despite the digestion model used, the fermented and fermented dried samples exhibited higher Mg content than the unfermented samples in the 3% to 30% range. However, a notable difference was only observed in Castellana lentils and white quinoa. The fermented and fermented dried samples of Castellana lentil and white and black quinoa exhibited a significant increase in Ca content ranging from 18% to 124%. However, the fermented and fermented dried samples of Pardina lentil exhibited a decrease in Ca content. Regarding Fe content, both Castellana and Pardina lentils experienced a notable increase, ranging from 63% to 329% in their fermented and fermented dried samples. Furthermore, a decrease was observed in the fermented and fermented dried samples of white and black quinoa. Therefore, it can be inferred that mineral bioaccessibility is enhanced through the fermentation process.

When comparing digestion models, it was found that the older adult digestion model demonstrated a decrease in mineral content when evaluated against the standard. Despite individual cases of significant differences, no overall significant differences were observed between the digestion models.

Many legume grains, cereals, and pseudocereal seeds contain varying concentrations of phytic acid. Upon ingestion, it remains undigested in the human digestive system due to the lack of the phytase enzyme. Phytic acid can bind to crucial micronutrients such as iron, calcium, magnesium, and zinc, reducing their absorption during gastrointestinal digestion [18]. Processes such as SSF are used to reduce this anti-nutrient. This study analyses the effects of SSF and drying on the bioaccessibility of phytic acid in unfermented, fermented, and fermented dried samples of Castellana and Pardina lentils and white and black quinoa. The analysis was conducted using standard and older adult in vitro digestion models, as illustrated in Figure 2. A marked reduction in the phytate content of approximately 90% can be observed in undigested fermented and fermented dried samples of Castellana lentil, as well as white and black quinoa as compared to their unfermented counterparts. Furthermore, there is no significant effect on Pardina lentils. These findings indicate that the decrease in this anti-nutritional factor is due to the activation of the endogenous phytase present in each substrate, as discussed previously [20,21]. When the undigested and digested samples were compared, a significant reduction in phytic acid release was observed after gastrointestinal digestion. The reductions ranged from 70% to 80% in unfermented lentil samples (Pardina and Castellana) and quinoa (white and black) regarding their initial content (undigested).

For both digested and fermented dried samples, black quinoa saw a reduction of approximately 40%, whereas Pardina lentils saw a reduction of approximately 80%. In contrast, the reduction in Castellana lentils and white quinoa was 100%, which was due to their minimal phytic acid content in undigested samples rather than the simulated gastrointestinal digestion’s physiological conditions. Therefore, a reduction in phytic acid in fermented and dehydrated fermented samples may be associated with increased levels of Mg, Ca, and Fe following gastrointestinal digestion. Chawla et al. [40] evaluated the impact of SSF in black-eyed pea seed flour using an *Aspergillus oryzae* strain on the mineral bioavailability of iron and zinc. They determined that after 96 h of fermentation, iron and zinc increased from 17.3% to 30.2% and from 14.4% to 29.6%, respectively. The authors attributed the improved mineral bioaccessibility to the degradation of anti-nutrient compounds, including phytic acid.

When comparing digestion models, a significant difference was observed between the standard model and the older adult model for Pardina lentils. However, no significant differences were found for Castellana lentils, white quinoa, and black quinoa. Therefore, there is no significant effect of the occurrence of digestive disorders with age. Couzy et al. [41] studied zinc absorption in older and younger subjects (with similar zinc status) using serum concentration curve (SCC). They administered soy milk fortified with 50 mg of zinc containing three levels of phytic acid: 0, 0.13, and 0.26 g/200 mL. They found that phytic acid reduced zinc absorption as the concentration of phytic acid increased in the beverage. Furthermore, they indicated that there were no differences between the older and younger subjects.

## 3. Materials and Methods

### 3.1. Materials

Lentils (*Lens culinaris*) of the Pardina and Castellana varieties from Hacendado^®^ and quinoa (*Chenopodium quinoa Wild*) of white and black varieties from the Hacendado^®^ and Nut&me brands, respectively, were obtained from local stores in Valencia (Spain). The *Pleurotus ostreatus* strain was acquired from the Spanish Type Culture Collection (CECT20311).

Pepsin from porcine gastric mucosa (≥3200 U/mg), pancreatin from porcine pancreas (8 × USP), bovine bile (dried, unfractionated), p-toluene-sulfonyl-L-arginine methyl ester (TAME, T4626), analytical grade salts (potassium chloride, potassium dihydrogen phosphate, sodium bicarbonate, sodium chloride, magnesium chloride hexahydrate, ammonium carbonate, and calcium chloride), potato starch, sodium phosphate, maltose standard, 3,5-dinitrosalicylic acid (DNS), potassium sodium tartrate tetrahydrate, sodium hydroxide, thioglycolic acid, phytic acid sodium salt hydrated from rice, 2,2’-bipyridine, formic acid, 2,2-diphenyl-1-picrylhydrazyl (DPPH), 2,2′-azino-bis (3-ethylbenzothiazoline-6-sulphonic acid) (ABTS), Folin–Ciocalteu reagent, 2,4,6-tripyridyl-s-triazine (TPTZ), gallic acid, (±)-6-Hydroxy-2,5,7,8-tetramethylchromane-2-carboxylic acid (Trolox), glucose, and mycopeptone were obtained from Sigma-Aldrich Co. (St. Louis, MO, USA).

For HPLC analysis, vanillic acid, quercetin 3-glucoside, quercetin, quercitrin, 4-hydroxybenzoic acid, rutin, epicatechin, trans-cinnamic acid, ferulic acid, naringenin, caffeic acid, 4-O-caffeoylquinic, p-coumaric acid, apigenin-7-glucoside, kaempferol, and sinapic acid were obtained from Sigma-Aldrich as an analytical standard (HPLC grade). Acetic acid glacial, concentrated hydrochloric acid, ethanol absolute, sodium carbonate, and ammonium iron (III) sulphate dodecahydrate were obtained from Panreac AppliChem (Barcelona, Spain). Acetonitrile (HPLC grade), methanol (HPLC grade), iron (III) chloride hexahydrate, sodium acetate trihydrate, and potassium persulphate were obtained from Honeywell Fluka (Morris Plains, NJ, USA). The malt extract and agar were obtained from Scharlau (Barcelona, Spain).

### 3.2. Fungal Solid-State Fermentation (SSF) and Flour Production

To perform SSF, a starter culture was first prepared by growing *P. ostreatus* mycelium in a Petri dish containing lentil or quinoa grains/seeds. The fermentation was then conducted by inoculating a portion of the starter culture into a glass jar containing 35 g of lentil or quinoa grains/seeds following the methodology used previously [20]. Fermented lentils and quinoa grains/seeds were dried using hot air drying and freeze drying methods, the latter used as the reference drying method because it was expected to have the best preservation of the sample properties according to literature. Hot air drying was conducted using a convective dryer (Pol-Eko-Aparatura, CLW 750 TOP+, Kokoszycka, Poland) at 70 °C with an air rate of 10.5 ± 0.2 m/s and an air humidity of 8.7 ± 1.2%. The samples were dried for 3.5 to 4 h to a target product moisture of 7% (wet basis). Freeze drying was performed using a freeze dryer (Telstar, Lyoquest-55, Terrassa, Spain) at −45 °C and 0.8 mBar for 48 h. Unfermented and fermented dried samples were then milled using a food processor (Vorwerk, Thermomix^®^ TM6-1, Wuppertal, Germany), applying 10,000 rpm at 15 s intervals for 1 min.

### 3.3. Simulated In Vitro Gastrointestinal Digestion under Standard and Older Adult Conditions

Unfermented, fermented, and fermented dried samples were digested under two static in vitro digestion models: the older adult model [42] and the healthy adult model (standard; as a control) [43,44] (Table 7). Enzymatic activities were determined before each experiment according to the supplementary information in the protocol published by Brodkorb et al. [43]. Simulated salivary (SSF), gastric (SGF), and intestinal (SIF) fluid were prepared daily for the standard and older adult digestion model considering the concentrations of enzymes, bile salts, and pH of each digestive phase.

To perform the oral stage, 5 g of sample was mixed with 5 mL of SSF containing the enzyme concentration according to the digestion model (Table 7), with adjusted pH, mixed at 25 rpm using an Intelli-Mixer RM-2 (Elmi Ltd., Riga, LV-1006, Latvia), and incubated in a thermostatic chamber (J.P. Selecta SA, Barcelona, Spain) at 37 °C for 2 min.

For the gastric stage, 10 mL of SGF was added to the food bolus according to the conditions simulated in each model (Table 7), the pH, mixed at 55 rpm, and incubated at 37 °C for 2 h. For the intestinal stage, 20 mL of SIF was added to the gastric chyme according to the concentration of enzyme and bile salts (Table 7), adjusted the pH, mixed at 55 rpm, and incubated at 37 °C for 2 h. After gastrointestinal digestion, enzyme activity was inhibited by adjusting the pH to 5 and keeping the samples in an ice bath. Finally, the samples were centrifuged at 8000× *g* for 10 min and aliquots of the bioaccessible fraction were taken for analytical determinations.

### 3.4. Analytical Determinations

#### 3.4.1. Total Phenolic Content (TPC)

TPC of the samples before and after undergoing in vitro digestion was determined using the Folin–Ciocalteu methodology as outlined by Chang et al. [45]. For the samples that were not digested, phenolic compounds were extracted by blending 2.5 g of the sample with 7.5 mL of the extraction solvent (a mixture of double distilled water and ethanol at 70:30) and adjusting the pH to 2 with 2 M HCl. The mixture was then treated to an ultrasonic bath (J.P. Selecta, 3000840) at 30 °C for 2 h. The pH was adjusted to 2 with 2 M HCl. The samples were centrifuged at 8000× *g* for 15 min, and the extraction process was repeated twice, with subsequent mixing of both extracted samples. The bioaccessible fraction determined the digested samples. An aliquot of 125 µL of the extract/digest was taken and mixed with 500 µL of bidistilled water, followed by 125 µL of the Folin–Ciocalteu reagent. This was left to react for 6 min. Then, 1.25 mL of 7% sodium carbonate and 1 mL of bidistilled water were added. The sample was incubated for 30 min at room temperature in darkness. Afterward, the absorbance was measured at 760 nm, and the results were presented in mg gallic acid/g dry basis using a standard curve.

#### 3.4.2. Antioxidant Activity

The antioxidant activity of the samples before and after in vitro digestion was determined by three methods: ABTS, DPPH, and FRAP, following the methodology described by Thaipong et al. [46]. The same extracts used in the TPC section were used for undigested and digested samples.

For the ABTS test, the working solution (7.4 mM ABTS and 2.6 mM potassium persulphate in a 1:1 ratio) was allowed to react for 12 h at room temperature in darkness. The working solution (1 mL) was diluted with methanol to obtain an absorbance close to 1.1 at 734 nm. Extract/digest (150 µL) was reacted with 2.85 mL of ABTS working solution for 2 h in darkness and absorbance was measured at 734 nm.

For the DPPH test, a fresh working solution of 0.039 g/L DPPH was prepared in pure methanol to obtain an absorbance close to 1.1 at 515 nm. Extract/digest (75 µL) reacted with 2.925 mL of DPPH working solution for 30 min in darkness and absorbance was measured at 515 nm.

For the FRAP test, fresh working solution was prepared by mixing 300 mM acetate buffer (3.1 g sodium acetate trihydrate and 16 mL acetic acid glacial in 1 L water, pH 3.6), TPTZ solution (10 mM 2,4,6-tripyridyl-s-triazine dissolved in 40 mM HCl), and 20 mM iron (III) chloride hexahydrate solution in a 10:1:1 ratio, respectively, and incubated at 37 °C before use. Extract/digest (150 µL) reacted with 2.85 mL of FRAP working solution for 30 min in darkness, and the absorbance was measured at 593 nm. The results are expressed as mg Trolox/g dry basis using a standard curve for the three antioxidant determination methods.

The antioxidant index was calculated for each sample for all antioxidant activity assays (ABTS, DPPH, and FRAP). An antioxidant index value of 100 was assigned to the highest sample score in each assay. Then, the antioxidant index was calculated for the entire group of samples in each assay according to Equation (1) [47]:(1)$$Antioxidant\;index\;(\%)=\left(\frac{sample\;score}{highest\;sample\;score}\right)\times 100$$

The overall APCI was calculated by averaging each sample’s antioxidant index (%) of each antioxidant activity assay.

#### 3.4.3. Phenolic Profile by HPLC Analysis

The phenolic profile of the samples after in vitro digestion was determined by filtering the bioaccessible fraction of the digest with a 0.45 µm PTFE filter. The samples were analysed using an HPLC 1200 Series Rapid Resolution coupled to a diode detector Serie (Agilent, Palo Alto, CA, USA) according to the methodology described by Tanleque-Alberto et al. [48]. A Brisa-LC 5 µm C18 column (250 × 4.6 mm) (Teknokroma, Barcelona, Spain) was used. Mobile phase A was 1% formic acid, and mobile phase B was acetonitrile (ACN). The following gradient program was used: 0 min, 90% A; 25 min, 40% A; 26 min, 20% A; held for 30 min; 35 min, 90% A; held for 40 min. Flow rate, injection volume, and working temperature of the column was 0.5 mL/min, 10 µL, and 30 °C, respectively. Unknown compounds were identified by comparing chromatographic retention times with reference standards according to the following wavelengths for each compound: 250 nm for vanillic acid; 260 nm for 4-hydroxybenzoic acid, rutin, quercetin 3-glucoside, and quercitrin; 280 nm for gallic acid, epicatechin, quercetin and trans-cinnamic acid; 290 nm for naringenin; 320 nm for 4-O-caffeoylquinic, caffeic acid, p-coumaric acid, sinapic acid, ferulic acid, and apigenin-7-glucoside; and 380 nm for kaempferol. The results are expressed as µg/g dry basis using a standard curve.

#### 3.4.4. Phytic Acid Content

The phytic acid content was measured before and after in vitro digestion following the protocol described by Haug and Lantzsch [49] and modified by Peng et al. [50]. For undigested samples, the extract was prepared by mixing 50 mg of sample with 10 mL of 0.2 M HCl and left overnight at 4 °C. For digested samples, the determination was performed on the bioaccessible fraction. An aliquot of 500 µL of the extract/digest was taken, and 1 mL of ferric solution (0.2 g of ammonium iron (III) sulphate dodecahydrate dissolved in 100 mL of 2 M hydrochloric acid and made up to 1 L with distilled water) was added. It was incubated in a boiling water bath for 30 min and then cooled to room temperature. The sample was centrifuged for 30 min at 3000× *g*, and 1 mL of the supernatant was taken and mixed with 1.5 mL of 2,2’-bipyridine solution (10 g of 2,2’-bipyridine and 10 mL of thioglycolic acid dissolved in distilled water and made up to 1 L). The results are expressed as mg phytic acid/g dry basis using a standard curve made with a stock solution of 1.3 mg/mL phytic acid concentration and diluted with 0.2 M hydrochloric acid between 0.1 and 1 mL (3.16–31.6 µg/mL phytate phosphorus).

#### 3.4.5. Mineral Quantification

The quantification of minerals (Fe, Ca, and Mg) before and after gastrointestinal digestion was performed by inductively coupled plasma mass spectrometry (ICP-MS). The mineral extract was prepared according to the methodology published by Barrera et al. [51]. A 5 g sample was weighed for undigested food and a 3.5 mL aliquot was taken from the bioaccessible fraction of digested food. The samples were incinerated at 600 °C for 10 h. The ashes were dissolved with 1 mL of 69% nitric acid and re-incinerated until white ashes were obtained. The white ashes were suspended in 1.5 mL of 69% nitric acid and 4 mL of bidistilled water.

The samples were analysed using an ICP-MS equipped with an autosampler (iCAP Q, Thermo, Waltham, MA, USA) according to the methodology proposed by Chen et al. [52]. The conditions of the working equipment were radio frequency power (1550 W), cool gas flow (14 L/min), auxiliary gas flow (0.8 L/min), nebuliser gas flow (1.08 L/min), peristaltic pump speed (40 rpm), sampling depth (5 mm), spray chamber temperature (2.7 °C), and dwell time (20 ms). The results are expressed as µg/g dry basis.

### 3.5. Statistical Analysis

The experiments were conducted at least in triplicate and results reported as mean ± standard deviation. A one-way ANOVA with a 95% confidence interval (*p* < 0.05) was performed to determine the statistical significance of the variables studied (SSF, drying, and common GI conditions of older adults) on the antioxidant activity, phenolic, phytates, and mineral contents in the bioaccessible fraction in lentil and quinoa samples, employing Statgraphics Centurion versionXV as statistical software.

## 4. Conclusions

SSF, coupled with drying at 70 °C, had a positive impact on the bioaccessibility of phenolic compounds and antioxidant activity, albeit to various degrees depending on the substrate. The profile of phenolic compounds following gastrointestinal digestion showed an increase in vanillic and caffeic acids in Castellana lentils and in vanillic acid in Pardina lentils, reaching approximately three times the levels of the unfermented samples. There was a significant increase in gallic acid of up to 7 and 1.4 times more than in the unfermented analogue in white and black quinoa, respectively. Regarding antioxidant activity, the Castellana lentil and white quinoa flours fermented and dried at 70 °C showed the highest APCI (>90% and >80%, respectively) after digestion, thus having a higher capacity to neutralise free radicals than the other samples. Fermented and fermented dried samples (at 70 °C and lyophilised) displayed a mineral bioaccessible content that was higher than the unfermented samples. This, together with the low phytic acid content present in fermented dried samples, renders such flours attractive for developing functional products with superior bioaccessibility than unfermented flours. Finally, typical age-related digestive conditions did not appear to affect the mineral bioavailability of Fe, Mg, and Ca in lentils and quinoa flours. However, these conditions reduced the phenolic profile and antioxidant activity of digesta when compared to the results obtained in the standard model.

Fermented Castellana and white quinoa flours are the optimal choice for product development, specifically catering to this population group to maximise health benefits. Furthermore, it is essential to evaluate the techno-functional properties of fermented flours to determine their compatibility with different food applications. Moreover, it is crucial to perform scale-up tests of the fermentation process to facilitate the technological transfer of this process to the food industry.

## Figures and Tables

**Figure 1 molecules-28-07298-f001:**
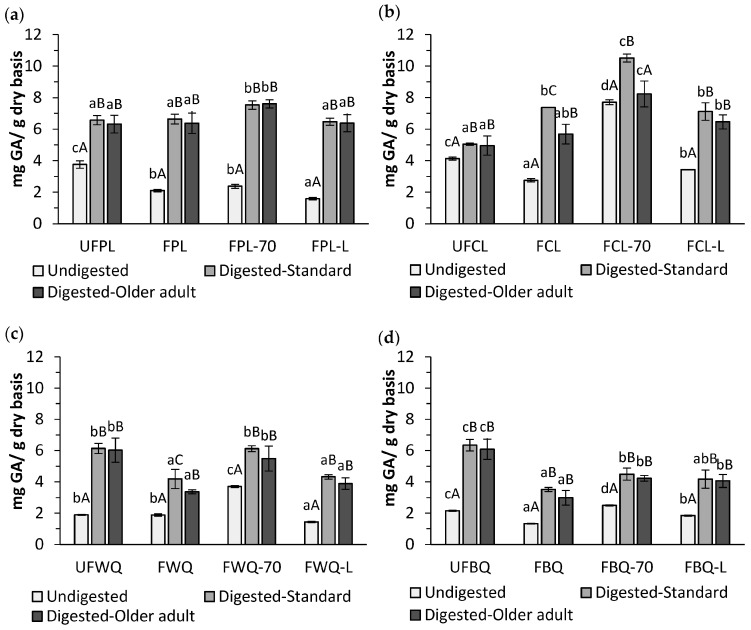
Total phenol content (mg gallic acid/g dry basis) in Pardina (**a**), Castellana lentil (**b**), white (**c**), and black quinoa (**d**) for unfermented flour (UFPL, UFCL, UFWQ, and UFBQ), fermented grain/seed (FPL, FCL, FWQ, and FBQ), fermented dried at 70 °C (FPL-70, FCL-70, FWQ-70, and FBQ-70), and fermented lyophilised (FPL-L, FCL-L, FWQ-L, and FBQ-L) flour obtained with a standard or older adult in vitro digestion model. ^a,b,c,d^ Different lowercase letters indicate significant differences (*p* < 0.05) between samples. ^A,B,C^ Different capital letters indicate significant differences (*p* < 0.05) between digestion models.

**Figure 2 molecules-28-07298-f002:**
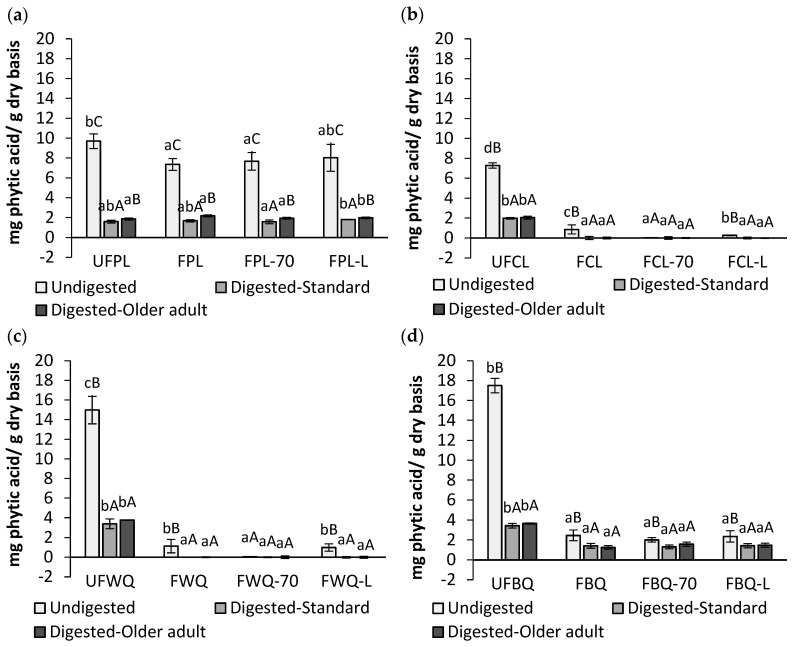
Phytic acid content (mg/g dry basis) in Pardina lentil (**a**) and Castellana lentil (**b**) and white (**c**) and black quinoa (**d**) for unfermented flour (UFPL, UFCL, UFWQ, and UFBQ), fermented grain/seed (FPL, FCL, FWQ, and FBQ), fermented dried at 70 °C (FPL-70, FCL-70, FWQ-70, and FBQ-70), and fermented lyophilised (FPL-L, FCL-L, FWQ-L, and FBQ-L) flour obtained with a standard and older adult in vitro digestion model. ^a,b,c^ Different lowercase letters indicate significant differences (*p* < 0.05) between samples. ^A,B,C^ Different capital letters indicate significant differences (*p* < 0.05) between digestion models.

**Table 1 molecules-28-07298-t001:** Phenolic content (µg/g dry basis) in digested Pardina lentil for unfermented flour (UFPL), fermented grain (FPL), fermented dried at 70 °C (FPL-70), and fermented lyophilised (FPL-L) flour.

	Digested (Standard)				Digested (Older Adult)			
	UFPL	FPL	FPL-70	FPL-L	UFPL	FPL	FPL-70	FPL-L
Phenolic acids								
Gallic acid	n.d.	n.d.	n.d.	n.d.	n.d.	n.d.	n.d.	n.d.
Caffeic acid	n.d.	n.d.	n.d.	n.d.	n.d.	n.d.	n.d.	n.d.
p-Coumaric acid	4.5 ± 0.5 ^cB^	2.87 ± 0.08 ^bB^	3.19 ± 0.19 ^bA^	2.14 ± 0.07 ^aA^	2.2 ± 0.2 ^aA^	2.33 ± 0.04 ^aA^	3.02 ± 0.18 ^bA^	2.0 ± 0.3 ^aA^
Sinapic acid	n.d.	n.d.	n.d.	n.d.	n.d.	n.d.	n.d.	n.d.
4-O-Caffeoylquinic	88 ± 31 ^bA^	49.5 ± 0.2 ^aB^	42 ± 3 ^aB^	60.6 ± 0.3 ^bB^	86 ± 7 ^cA^	34 ± 5 ^aA^	32 ± 3 ^aA^	46.9 ± 0.3 ^bA^
4-Hydroxybenzoic acid	4.4 ± 0.2 ^aA^	4.14 ± 0.15 ^aB^	4.0 ± 0.5 ^aB^	3.88 ± 0.13 ^aB^	3.3 ± 0.3 ^bA^	3.375 ± 0.010 ^bA^	2.1 ± 0.3 ^aA^	2.90 ± 0.17 ^bA^
Vanillic acid	7.6 ± 0.5 ^aA^	19.8 ± 0.6 ^cB^	20.7 ± 0.5 ^cB^	17.81 ± 0.12 ^bB^	7.5 ± 0.7 ^aA^	13.89 ± 0.12 ^bA^	16.8 ± 0.7 ^cA^	13.4 ± 1.3 ^bA^
Ferulic acid	4.6 ± 1.2 ^bA^	2.29 ± 0.02 ^aB^	2.1 ± 1.3 ^aA^	2.58 ± 0.09 ^aA^	3.0 ± 0.2 ^cA^	2.03 ± 0.03 ^aA^	2.06 ± 0.03 ^aA^	2.47 ± 0.11 ^bA^
trans-Cinnamic acid	traces	3.02 ± 0.08 ^aA^	9.5 ± 0.7 ^bA^	2.19 ± 0.17 ^aA^	n.d.	2.75 ± 0.02 ^aA^	9.0 ± 0.6 ^bA^	2.0 ± 0.3 ^aA^
Flavonoids								
Rutin	n.d.	n.d.	n.d.	n.d.	n.d.	n.d.	n.d.	n.d.
Epicatechin	n.d.	n.d.	n.d.	n.d.	n.d.	n.d.	n.d.	n.d.
Quercetin 3-glucoside	n.d.	n.d.	n.d.	n.d.	n.d.	n.d.	n.d.	n.d.
Quercitrin	4.7 ± 0.4 ^B^	n.d.	traces	traces	2.0 ± 0.4 ^A^	traces	traces	traces
Apigenin-7-glucoside	0.62 ± 0.07 ^A^	n.d.	traces	traces	2.80 ± 0.06 ^cB^	2.28 ± 0.03 ^b^	2.195 ± 0.007 ^a^	2.67 ± 0.10 ^c^
Quercetin	5.3 ± 0.4 ^aB^	5.181 ± 0.006 ^a^	5.0 ± 0.4 ^a^	5.776 ± 0.008 ^a^	0.78 ± 0.05 ^A^	traces	traces	traces
Naringenin	n.d.	n.d.	n.d.	n.d.	n.d.	n.d.	n.d.	n.d.
Kaempferol	n.d.	n.d.	n.d.	n.d.	n.d.	n.d.	n.d.	n.d.

The results represent the mean of three repetitions with their standard deviation. ^a,b,c^ Different lowercase letters indicate significant differences between flours, and ^A,B^ different capital letters indicate significant differences between digestion models (*p* < 0.05); n.d.: not detected; and traces: not quantifiable.

**Table 2 molecules-28-07298-t002:** Phenolic content (µg/g dry basis) in digested Castellana lentil for unfermented flour (UFCL), fermented grain (FCL), fermented dried at 70 °C (FCL-70), and fermented lyophilised (FCL-L) flour.

	Digested (Standard)				Digested (Older Adult)			
	UFCL	FCL	FCL-70	FCL-L	UFCL	FCL	FCL-70	FCL-L
Phenolic acids								
Gallic acid	n.d.	n.d.	n.d.	n.d.	n.d.	n.d.	n.d.	n.d.
Caffeic acid	3.4 ± 0.4 ^aB^	5.5 ± 0.7 ^bB^	10.8 ± 1.3 ^cB^	5.9 ± 0.4 ^bA^	2.50 ± 0.06 ^aA^	3.3 ± 0.2 ^abA^	4.4 ± 0.2 ^bA^	6.6 ± 0.7 ^cB^
p-Coumaric acid	6.2 ± 1.0 ^cA^	1.91 ± 0.04 ^aB^	3.0 ± 0.2 ^bB^	1.74 ± 0.05 ^aA^	6.4 ± 1.3 ^bA^	1.637 ± 0.007 ^aA^	1.80 ± 0.14 ^aA^	1.86 ± 0.08 ^aA^
Sinapic acid	n.d.	n.d.	n.d.	n.d.	n.d.	n.d.	n.d.	n.d.
4-O-Caffeoylquinic	n.d.	n.d.	n.d.	n.d.	n.d.	n.d.	n.d.	n.d.
4-Hydroxybenzoic acid	n.d.	4.5 ± 1.4 ^a^	5.5 ± 0.9 ^a^	7.4 ± 0.4 ^a^	n.d.	n.d.	n.d.	n.d.
Vanillic acid	6.2 ± 1.2 ^a^	18 ± 3 ^b^	20 ± 2 ^b^	24 ± 2 ^b^	traces	traces	traces	traces
Ferulic acid	traces	n.d.	n.d.	n.d.	n.d.	n.d.	n.d.	n.d.
trans-Cinnamic acid	n.d.	1.6 ± 0.3 ^aA^	8.9 ± 0.9 ^bB^	1.93 ± 0.03 ^aA^	n.d.	2.25 ± 0.06 ^aB^	5.8 ± 0.2 ^cA^	3.3 ± 0.3 ^bB^
Flavonoids								
Rutin	n.d.	n.d.	n.d.	n.d.	n.d.	n.d.	n.d.	n.d.
Epicatechin	n.d.	n.d.	n.d.	n.d.	n.d.	n.d.	n.d.	n.d.
Quercetin 3-glucoside	traces	traces	traces	n.d.	traces	n.d.	n.d.	n.d.
Quercitrin	n.d.	n.d.	n.d.	n.d.	n.d.	n.d.	n.d.	n.d.
Apigenin-7-glucoside	n.d.	2.61 ± 0.09 ^aB^	5.4 ± 0.6 ^bB^	3.23 ± 0.06 ^aA^	n.d.	2.33 ± 0.03 ^aA^	3.8 ± 0.4 ^cA^	3.13 ± 0.15 ^bA^
Quercetin	n.d.	n.d.	n.d.	n.d.	traces	traces	traces	traces
Naringenin	n.d.	traces	traces	traces	n.d.	traces	traces	traces
Kaempferol	n.d.	n.d.	n.d.	n.d.	n.d.	n.d.	n.d.	n.d.

The results represent the mean of three repetitions with their standard deviation. ^a,b,c^ Different lowercase letters indicate significant differences between flours, and ^A,B^ different capital letters indicate significant differences between digestion models (*p* < 0.05); n.d.: not detected; and traces: not quantifiable.

**Table 3 molecules-28-07298-t003:** Phenolic content (µg/g dry basis) in digested white quinoa for unfermented flour (UFWQ), fermented seeds (FWQ), fermented dried at 70 °C (FWQ-70), and fermented lyophilised (FWQ-L) flour.

	Digested (Standard)				Digested (Older Adult)			
	UFWQ	FWQ	FWQ-70	FWQ-L	UFWQ	FWQ	FWQ-70	FWQ-L
Phenolic acids								
Gallic acid	20 ± 2 ^a^	77 ± 7 ^cA^	139 ± 13 ^dB^	56 ± 3 ^bA^	traces	68 ± 8 ^bA^	75 ± 9 ^bA^	43 ± 7 ^aA^
Caffeic acid	6.7 ± 0.9 ^b^	2.09 ± 0.09 ^aB^	2.7 ± 0.6 ^aB^	traces	traces	0.83 ± 0.06 ^aA^	0.88 ± 0.02 ^aA^	traces
p-Coumaric acid	2.9 ± 0.9 ^A^	n.d.	traces	traces	6.8 ± 1.0 ^B^	n.d.	traces	traces
Sinapic acid	n.d.	n.d.	n.d.	n.d.	n.d.	n.d.	n.d.	n.d.
4-O-Caffeoylquinic	n.d.	n.d.	n.d.	n.d.	n.d.	n.d.	n.d.	n.d.
4-Hydroxybenzoic acid	n.d.	traces	n.d.	n.d.	n.d.	traces	n.d.	n.d.
Vanillic acid	16 ± 2 ^bB^	3.5 ± 0.4 ^aA^	5.0 ± 0.4 ^aA^	3.1 ± 0.3 ^aA^	8.3 ± 0.9 ^cA^	2.93 ± 0.02 ^aA^	4.1448 ± 0.0014 ^bA^	3.0 ± 0.4 ^aA^
Ferulic acid	9.0 ± 1.0 ^bA^	traces	1.68 ± 0.12 ^aA^	traces	8.6 ± 0.7 ^bA^	traces	1.699 ± 0.004 ^aA^	traces
trans-Cinnamic acid	traces	1.3 ± 0.2 ^bA^	2.95 ± 0.05 ^cA^	0.74 ± 0.05 ^aA^	traces	2.14 ± 0.08 ^bB^	3.69 ± 0.02 ^cB^	1.87 ± 0.10 ^aB^
Flavonoids								
Rutin	n.d.	n.d.	n.d.	n.d.	n.d.	n.d.	n.d.	n.d.
Epicatechin	n.d.	n.d.	n.d.	n.d.	n.d.	n.d.	n.d.	n.d.
Quercetin 3-glucoside	4.5 ± 1.0 ^a^	3.162 ± 0.014 ^a^	n.d.	n.d.	n.d.	n.d.	n.d.	n.d.
Quercitrin	6.2 ± 0.3 ^abA^	6.7 ± 0.9 ^bB^	6.60 ± 0.08 ^abA^	5.3 ± 0.4 ^aA^	6.8 ± 1.7 ^bA^	3.51 ± 0.10 ^aA^	5.9 ± 0.7 ^abA^	4.0 ± 1.1 ^abA^
Apigenin-7-glucoside	2.23 ± 0.04 ^a^	2.62 ± 0.08 ^aA^	9.3 ± 0.4 ^cB^	3.43 ± 0.14 ^bB^	n.d.	2.33 ± 0.03 ^bA^	8.40 ± 0.03 ^cA^	1.99 ± 0.04 ^aA^
Quercetin	traces	traces	traces	traces	traces	traces	traces	traces
Naringenin	n.d.	n.d.	n.d.	n.d.	n.d.	n.d.	n.d.	n.d.
Kaempferol	n.d.	n.d.	n.d.	n.d.	n.d.	n.d.	n.d.	n.d.

The results represent the mean of three repetitions with their standard deviation. ^a,b,c,d^ Different lowercase letters indicate significant differences between flours, and ^A,B^ different capital letters indicate significant differences between digestion models (*p* < 0.05); n.d.: not detected; and traces: not quantifiable.

**Table 4 molecules-28-07298-t004:** Phenolic content (µg/g dry basis) in digested black quinoa for unfermented flour (UFBQ), fermented seed (FBQ), fermented dried at 70 °C (FBQ-70), and fermented lyophilised (FBQ-L) flour.

	Digested (Standard)				Digested (Older Adult)			
	UFBQ	FBQ	FBQ-70	FBQ-L	UFBQ	FBQ	FBQ-70	FBQ-L
Phenolic acids								
Gallic acid	30 ± 4 ^aB^	21 ± 3 ^aB^	42 ± 2 ^bB^	39 ± 3 ^bB^	15 ± 2 ^aA^	16.03 ± 0.18 ^aA^	29.81 ± 0.12 ^bA^	27.7 ± 2.0 ^bA^
Caffeic acid	2.08 ± 0.18 ^A^	n.d.	n.d.	n.d.	1.87 ± 0.06 ^A^	n.d.	n.d.	n.d.
p-Coumaric acid	traces	traces	traces	traces	1.97 ± 0.06 ^b^	traces	traces	1.30 ± 0.07 ^a^
Sinapic acid	15 ± 3 ^bA^	1.29 ± 0.17 ^aA^	traces	2.19 ± 0.03 ^aB^	32 ± 3 ^bB^	1.34 ± 0.11 ^aA^	traces	1.64 ± 0.12 ^aA^
4-O-Caffeoylquinic	n.d.	n.d.	n.d.	n.d.	n.d.	n.d.	n.d.	n.d.
4-Hydroxybenzoic acid	n.d.	n.d.	n.d.	n.d.	n.d.	n.d.	n.d.	n.d.
Vanillic acid	n.d.	n.d.	n.d.	n.d.	n.d.	n.d.	n.d.	n.d.
Ferulic acid	4.9 ± 0.4 ^A^	traces	traces	traces	8.79 ± 0.04 ^B^	traces	traces	traces
trans-Cinnamic acid	traces	traces	traces	traces	traces	traces	traces	traces
Flavonoids								
Rutin	n.d.	n.d.	n.d.	n.d.	n.d.	n.d.	n.d.	n.d.
Epicatechin	traces	n.d.	n.d.	n.d.	traces	traces	traces	traces
Quercetin 3-glucoside	n.d.	n.d.	n.d.	n.d.	n.d.	n.d.	n.d.	n.d.
Quercitrin	10.0 ± 1.0 ^bA^	3.7 ± 0.4 ^aA^	5 ± 3 ^abA^	8.71 ± 0.11 ^abB^	11.5 ± 1.3 ^bA^	6.95 ± 0.07 ^aB^	6.030 ± 0.009 ^aA^	5.4 ± 0.3 ^aA^
Apigenin-7-glucoside	n.d.	2.23 ± 0.13 ^aA^	3.6 ± 0.5 ^bA^	3.92 ± 0.02 ^bB^	n.d.	3.0 ± 0.3 ^aA^	3.2 ± 0.4 ^aA^	2.9 ± 0.3 ^aA^
Quercetin	n.d.	n.d.	n.d.	n.d.	n.d.	n.d.	n.d.	n.d.
Naringenin	n.d.	1.58 ± 0.18 ^aA^	2.6 ± 0.3 ^bA^	3.04 ± 0.08 ^bB^	n.d.	2.77 ± 0.08 ^aB^	2.33 ± 0.15 ^aA^	2.26 ± 0.10 ^aA^
Kaempferol	n.d.	n.d.	n.d.	n.d.	n.d.	n.d.	n.d.	n.d.

The results represent the mean of three repetitions with their standard deviation. ^a,b^ Different lowercase letters indicate significant differences between flours, and ^A,B^ different capital letters indicate significant differences between digestion models (*p* < 0.05); n.d.: not detected; and traces: not quantifiable.

**Table 5 molecules-28-07298-t005:** Antioxidant activity (mg Trolox/g dry basis) by the ABTS, DPPH, and FRAP methods and total phenol content (mg gallic acid/g dry basis) in undigested and digested Pardina and Castellana lentils and white and black quinoa for unfermented flour (UFPL, UFCL, UFWQ, and UFBQ), fermented grain/seed (FPL, FCL, FWQ, and FBQ), fermented dried at 70 °C (FPL-70, FCL-70, FWQ-70, and FBQ-70), and fermented lyophilised (FPL-L, FCL-L, FWQ-L, and FBQ-L) flour, under standard and older adult simulated gastrointestinal conditions.

	Antioxidant Activity									APCI *		
	ABTS and ABTS Index			DPPH and DPPH Index			FRAP and FRAP Index					
	Undigested	Digested (Standard)	Digested (Older Adult)	Undigested	Digested (Standard)	Digested (Older Adult)	Undigested	Digested (Standard)	Digested (Older Adult)	Undig.	Standard	Older Adult
Pardina Lentil												
UFPL	9.5 ± 0.4 ^dA^ (100.0)	11.5 ± 0.3 ^aB^ (84.0)	11.6 ± 0.5 ^abB^ (92.3)	2.07 ± 0.09 ^cA^ (100.0)	2.29 ± 0.08 ^cB^ (100.0)	2.51 ± 0.09 ^cC^ (100.0)	7.6 ± 0.2 ^bB^ (100.0)	4.1 ± 0.4 ^cA^ (100.0)	3.79 ± 0.18 ^cA^ (100.0)	100.0	94.7	97.4
FPL	5.7 ± 0.5 ^cA^ (60.7)	12.8 ± 0.4 ^bC^ (93.2)	11.4 ± 0.2 ^abB^ (90.9)	0.64 ± 0.04 ^bC^ (30.8)	0.53 ± 0.04 ^aB^ (22.9)	0.44 ± 0.05 ^aA^ (17.5)	0.31 ± 0.02 ^aA^ (4.1)	1.8 ± 0.2 ^aC^ (43.2)	1.35 ± 0.11 ^aB^ (35.6)	31.9	53.1	48.0
FPL-70	3.91 ± 0.16 ^bA^ (41.4)	13.7 ± 0.5 ^bB^ (100.0)	12.5 ± 1.2 ^bB^ (100.0)	0.516 ± 0.010 ^aA^ (25.0)	0.88 ± 0.04 ^bC^ (38.4)	0.76 ± 0.06 ^bB^ (30.2)	0.351 ± 0.007 ^aA^ (4.6)	2.60 ± 0.14 ^bB^ (62.7)	2.38 ± 0.19 ^bB^ (62.8)	23.7	67.0	64.3
FPL-L	3.20 ± 0.04 ^aA^ (33.9)	11.8 ± 0.8 ^aB^ (85.7)	11.0 ± 0.5 ^aB^ (87.9)	0.502 ± 0.014 ^aA^ (24.3)	0.83 ± 0.05 ^bC^ (36.1)	0.70 ± 0.05 ^bB^ (27.8)	0.31 ± 0.02 ^aA^ (4.1)	2.54 ± 0.11 ^bB^ (61.3)	2.3 ± 0.2 ^bB^ (61.3)	20.7	61.0	59.0
Castellana Lentil												
UFCL	8.4 ± 0.4 ^cA^ (100.0)	14.0 ± 1.4 ^aB^ (80.7)	12.2 ± 1.5 ^aB^ (70.5)	1.634 ± 0.015 ^bA^ (72.0)	1.65 ± 0.05 ^cA^ (100.0)	1.78 ± 0.16 ^cA^ (100.0)	8.3 ± 0.2 ^dB^ (100.0)	3.2 ± 0.4 ^cA^ (70.2)	3.02 ± 0.06 ^cA^ (67.9)	90.7	83.6	79.5
FCL	2.50 ± 0.09 ^aA^ (29.9)	16.9 ± 1.3 ^abC^ (97.2)	14.3 ± 1.4 ^aB^ (82.7)	2.27 ± 0.13 ^cB^ (100.0)	0.25 ± 0.04 ^aA^ (15.0)	0.213 ± 0.017 ^aA^ (11.9)	1.10 ± 0.03 ^aC^ (13.4)	0.89 ± 0.02 ^aB^ (19.8)	0.77 ± 0.05 ^aA^ (17.4)	47.7	44.0	37.3
FCL-70	6.2 ± 0.2 ^bA^ (73.9)	17.4 ± 1.8 ^bB^ (100.0)	17.2 ± 1.5 ^bB^ (100.0)	1.71 ± 0.02 ^bA^ (75.1)	1.64 ± 0.10 ^cA^ (99.3)	1.55 ± 0.10 ^bA^ (87.1)	7.0 ± 0.3 ^cB^ (85.3)	4.50 ± 0.15 ^dA^ (100.0)	4.45 ± 0.11 ^dA^ (100.0)	78.1	99.8	95.7
FCL-L	2.32 ± 0.16 ^aA^ (27.7)	14.5 ± 1.7 ^abB^ (83.5)	13.7 ± 1.7 ^aB^ (79.5)	1.093 ± 0.016 ^aC^ (48.2)	0.39 ± 0.04 ^bB^ (23.4)	0.323 ± 0.014 ^aA^ (18.1)	2.14 ± 0.05 ^bC^ (25.9)	1.38 ± 0.07 ^bB^ (30.7)	1.13 ± 0.13 ^bA^ (25.3)	33.9	45.9	41.0
White Quinoa												
UFWQ	1.48 ± 0.08 ^bA^ (64.7)	8.20 ± 1.14 ^aB^ (64.5)	7.1 ± 0.6 ^aB^ (61.5)	1.070 ± 0.005 ^cC^ (80.2)	0.48 ± 0.04 ^cB^ (100.0)	0.23 ± 0.02 ^bA^ (75.0)	1.53 ± 0.05 ^cB^ (81.5)	1.22 ± 0.14 ^cA^ (100.0)	1.70 ± 0.03 ^cB^ (100.0)	75.5	88.2	78.8
FWQ	1.87 ± 0.08 ^cA^ (81.7)	10.1 ± 1.2 ^abB^ (79.4)	9.1 ± 1.2 ^aB^ (78.5)	1.334 ± 0.012 ^dB^ (100.0)	0.127 ± 0.008 ^aA^ (26.4)	0.119 ± 0.011 ^aA^ (38.5)	0.47 ± 0.03 ^aB^ (24.9)	0.34 ± 0.05 ^aA^ (27.8)	0.53 ± 0.07 ^aB^ (31.3)	68.9	44.5	49.5
FWQ-70	2.287 ± 0.006 ^dA^ (100.0)	12.7 ± 1.3 ^cB^ (100.0)	11.6 ± 1.8 ^bB^ (100.0)	0.80 ± 0.04 ^aB^ (60.3)	0.37 ± 0.02 ^bA^ (76.2)	0.31 ± 0.04 ^cA^ (100.0)	1.88 ± 0.09 ^dB^ (100.0)	1.05 ± 0.04 ^bA^ (86.2)	1.15 ± 0.16 ^bA^ (67.7)	86.8	87.5	89.2
FWQ-L	1.12 ± 0.04 ^aA^ (49.1)	10.8 ± 1.0b ^cC^ (84.5)	8.1 ± 0.8 ^aB^ (69.7)	1.00 ± 0.03 ^bB^ (74.8)	0.104 ± 0.009 ^aA^ (21.6)	0.093 ± 0.013 ^aA^ (30.3)	0.63 ± 0.03 ^bB^ (33.3)	0.40 ± 0.02 ^aA^ (32.5)	0.49 ± 0.08 ^aA^ (28.7)	52.4	46.2	42.9
Black Quinoa												
UFBQ	2.48 ± 0.03 ^cA^ (100.0)	10.1 ± 0.7 ^aC^ (99.0)	7.3 ± 0.6 ^bB^ (99.4)	0.88 ± 0.03 ^bB^ (63.6)	0.69 ± 0.04 ^dA^ (100.0)	0.63 ± 0.03 ^dA^ (100.0)	2.74 ± 0.04 ^dC^ (100.0)	1.7 ± 0.2 ^cA^ (100.0)	2.1 ± 0.3 ^bB^ (100.0)	87.9	99.7	99.8
FBQ	1.69 ± 0.10 ^bA^ (68.2)	9.6 ± 0.4 ^aC^ (94.1)	6.7 ± 0.7 ^abB^ (90.8)	1.38 ± 0.02 ^dB^ (100.0)	0.25 ± 0.02 ^bA^ (35.5)	0.26 ± 0.03 ^cA^ (40.9)	0.52 ± 0.02 ^aA^ (19.1)	0.42 ± 0.03 ^aA^ (25.1)	0.71 ± 0.09 ^aB^ (34.3)	62.4	51.6	55.3
FBQ-70	1.68 ± 0.06 ^bA^ (67.8)	10.2 ± 0.5 ^aC^ (100.0)	7.4 ± 0.2 ^bB^ (100.0)	0.666 ± 0.014 ^aC^ (48.3)	0.36 ± 0.06 ^cB^ (52.4)	0.20 ± 0.02 ^bA^ (31.0)	1.26 ± 0.04 ^cB^ (46.2)	0.66 ± 0.08 ^bA^ (39.8)	0.74 ± 0.05 ^aA^ (35.5)	54.1	64.1	55.5
FBQ-L	1.33 ± 0.05 ^aA^ (53.7)	9.8 ± 1.2 ^aC^ (96.1)	5.5 ± 1.0 ^aB^ (74.6)	0.96 ± 0.06 ^cB^ (69.6)	0.13 ± 0.02 ^aA^ (19.4)	0.139 ± 0.009 ^aA^ (22.0)	0.82 ± 0.05 ^bB^ (30.0)	0.49 ± 0.08 ^abA^ (29.7)	0.58 ± 0.09 ^aA^ (27.9)	51.1	48.4	41.5

The results represent the mean of three repetitions with their standard deviation. ^a,b,c,d^ Different lowercase letters indicate significant differences (*p* < 0.05) between flours. ^A,B,C^ Different capital letters indicate significant differences (*p* < 0.05) between digestion models. Values in parentheses correspond to ABTS, DPPH, and FRAP indexes, calculated among unfermented, fermented, and fermented dried at 70 °C, and lyophilised samples of each plant food. * APCI: antioxidant potency composite index.

**Table 6 molecules-28-07298-t006:** Mineral content (µg/g dry basis) in undigested and digested Pardina and Castellana lentils and white and black quinoa for unfermented flour (UFPL, UFCL, UFWQ, and UFBQ), fermented grain/seed (FPL, FCL, FWQ, and FBQ), fermented dried at 70 °C (FPL-70, FCL-70, FWQ-70, and FBQ-70), and fermented lyophilised (FPL-L, FCL-L, FWQ-L, and FBQ-L) flour, under standard and older adult simulated gastrointestinal conditions.

	Magnesium (Mg)			Calcium (Ca)			Iron (Fe)		
	Undigested	Digested (Standard)	Digested (Older Adult)	Undigested	Digested (Standard)	Digested (Older Adult)	Undigested	Digested (Standard)	Digested (Older Adult)
Pardina Lentil									
UFPL	112.6 ± 0.9 ^aB^	76 ± 4 ^aA^	71 ± 3 ^aA^	62 ± 2 ^aB^	43 ± 4 ^cA^	44.1 ± 0.6 ^bA^	11.3 ± 0.3 ^cB^	1.60 ± 0.05 ^aA^	1.05 ± 0.14 ^aA^
FPL	125 ± 3 ^bB^	78 ± 2 ^aA^	70 ± 3 ^aA^	90.5 ± 0.9 ^cB^	22 ± 3 ^bA^	15 ± 3 ^aA^	10.05 ± 0.04^bC^	3.168 ± 0.004 ^cB^	2.45 ± 0.02 ^bA^
FPL-70	112 ± 2 ^aB^	74 ± 5 ^aA^	75 ± 10 ^aA^	79.6 ± 0.4 ^bB^	12 ± 2 ^aA^	12 ± 4 ^aA^	8.92 ± 0.05 ^aB^	2.6 ± 0.3 ^bA^	2.2 ± 0.3 ^bA^
FPL-L	109 ± 2 ^aB^	71 ± 7 ^aA^	76.4 ± 0.6 ^aA^	80 ± 2 ^bB^	44 ± 5 ^cA^	52 ± 2 ^cA^	8.9 ± 0.3 ^aB^	2.60 ± 0.02 ^bA^	2.78 ± 0.08 ^cA^
Castellana Lentil									
UFCL	122 ± 2 ^aB^	82 ± 7 ^aA^	84.8 ± 0.9 ^aA^	64.1 ± 0.9 ^aB^	48 ± 6 ^aA^	43 ± 4 ^aA^	9.04 ± 0.07 ^aB^	1.4 ± 0.3 ^aA^	1.5 ± 0.3 ^aA^
FCL	127 ± 4 ^abB^	93 ± 5 ^abA^	96 ± 2 ^cA^	94.0 ± 1.0 ^bB^	61 ± 6 ^bA^	57 ± 9 ^bA^	8.9 ± 0.2 ^aC^	5.8 ± 1.0 ^bB^	4.43 ± 0.03 ^cA^
FCL-70	142 ± 4 ^cC^	102 ± 2 ^bB^	91 ± 2 ^bA^	102 ± 5 ^bB^	58 ± 3 ^bA^	43 ± 6 ^aA^	9.2 ± 0.3 ^aC^	5.0 ± 0.2 ^bB^	3.8 ± 0.2 ^bA^
FCL-L	135 ± 3 ^bcC^	107 ± 10 ^bB^	92.5 ± 0.4 ^bcA^	101 ± 6 ^bB^	49 ± 5 ^aA^	54 ± 5 ^abA^	8.8 ± 0.2 ^aB^	6.0 ± 1.4 ^bA^	4.56 ± 0.12 ^cA^
White Quinoa									
UFWQ	218 ± 2 ^aB^	146 ± 3 ^aA^	139 ± 6 ^abA^	63 ± 2 ^aB^	32.1 ± 1.5 ^aA^	38 ± 3 ^aA^	3.67 ± 0.02 ^bB^	3.3 ± 0.2 ^bA^	3.1 ± 0.6 ^aA^
FWQ	232 ± 10 ^aB^	163.7 ± 1.3 ^bA^	157 ± 10 ^bA^	81 ± 4 ^bB^	72 ± 7 ^cB^	47 ± 4 ^cA^	3.45 ± 0.02 ^aB^	2.4 ± 0.2 ^aA^	2.0 ± 0.5 ^aA^
FWQ-70	261 ± 3 ^bB^	163 ± 7 ^bA^	156 ± 3 ^bA^	79 ± 3 ^bC^	65 ± 6 ^cB^	45 ± 3 ^bA^	4.54 ± 0.07 ^cB^	3.50 ± 0.04 ^bA^	2.9 ± 0.5 ^aA^
FWQ-L	248.3 ± 1.3 ^bC^	166 ± 5 ^bB^	133 ± 11 ^aA^	78.4 ± 0.6 ^bB^	49.8 ± 0.7 ^bA^	54 ± 3 ^cA^	3.78 ± 0.13 ^bB^	2.9 ± 0.3 ^abA^	2.2 ± 0.6 ^aA^
Black Quinoa									
UFBQ	210 ± 8 ^aB^	111 ± 10 ^aA^	135 ± 6 ^aA^	55 ± 2 ^aB^	36 ± 4 ^aA^	36 ± 5 ^aA^	4.3 ± 0.3 ^abB^	2.9 ± 0.4 ^bA^	3.3 ± 0.2 ^bA^
FBQ	212.0 ± 1.4 ^aB^	136 ± 2 ^abA^	135 ± 11 ^aA^	63 ± 2 ^bA^	56 ± 5 ^bA^	60 ± 10 ^cA^	3.86 ± 0.03 ^aB^	1.9 ± 0.2 ^aA^	1.54 ± 0.09 ^aA^
FBQ-70	217 ± 12 ^aB^	145 ± 2 ^bA^	138.9 ± 0.7 ^aA^	62 ± 2 ^bA^	57 ± 12 ^bA^	41 ± 4 ^bA^	4.5 ± 0.2 ^bB^	2.2 ± 0.6 ^bA^	1.6 ± 0.3 ^aA^
FBQ-L	220 ± 7 ^aB^	130 ± 16 ^abA^	144 ± 2 ^aA^	66 ± 4 ^bA^	58 ± 10 ^bA^	42 ± 5 ^bA^	4.4 ± 0.2 ^abB^	1.7 ± 0.3 ^aA^	1.8 ± 0.3 ^aA^

The results represent the mean of three repetitions with their standard deviation. ^a,b,c^ Different lowercase letters indicate significant differences (*p* < 0.05) between flours. ^A,B,C^ Different capital letters indicate significant differences (*p* < 0.05) between digestion models.

**Table 7 molecules-28-07298-t007:** Gastrointestinal conditions established for an in vitro digestion model for a healthy adult (standard) [43,44] and an older adult [42].

Digestive Stage	Digestion Models	
	Healthy Adult (Standard)	Older Adult
Oral stage	Amylase (75 U/mL)	**Amylase (112.5 U/mL)**
	pH 7	pH 7
	2 min	2 min
Gastric stage	Pepsin (2000 U/mL)	**Pepsin (1200 U/mL)**
	pH 3	**pH 3.7**
	2 h	2h
Intestinal stage	Pancreatin (100 U/mL)	**Pancreatin (80 U/mL)**
	Bile salts (10 mM)	**Bile salts (7 mM)**
	pH 7	pH 7
	2 h	2 h

The alterations made to the model for older adults compared to the standard model are highlighted in bold text.

## Data Availability

Data is contained within the article.

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
