# Peer review of "In Vitro Digestion Assessment (Standard vs. Older Adult Model) on Antioxidant Properties and Mineral Bioaccessibility of Fermented Dried Lentils and Quinoa"

_molecules, 2023, doi:10.3390/molecules28217298_

Round 1

Reviewer 1 Report

Comments and Suggestions for Authors

The paper described the changes in polyhenols, mineral composition and antioxidant activity occurring in lentil and quinoa samples submitted to different treatments following the digestion process which has been simulted in standard and elderly conditions.

It is interesting because the need to find value-added products for elderly is a real key problem of these and future years.

However, the manuscript need a revision according the following suggestions:

Abstract

“in Castellana 21 lentils from 5.05-10.5 mg gallic acid/g dry basis and in Pardina lentils from 6.6-7.5 mg gallic acid/g…”: from…to what?

“APCI” avoid to use abbreviation without before explaining its meaning

Results and discussion

Line 104 “highest” and not “higher”

Lines 120-122: considering that only one lentil sample shows a significative difference between older adult digestive and standard conditions, what stated by the authors is excessive. In addition for white quinoa no significative difference is present. The sentence sgould be revised.

Line 140: change “characterizarion”. It is no the right word

Line 155: “bioaccessible” and not “bio accessible”

Lines 182-189: this part is not well linked to the below part od the section because the bioactivity tested is the antioxidant one

Lines 277-292: delete because it is a repetition

Add the chromatographic profile of a lentil o quinoa samples in the different treatment conditions in Supplementary material

English language revisioni is suggested

Comments on the Quality of English Language

Minor English revision is needed

Author Response

ANSWER TO REVIEWERS

The authors are very grateful for the reviewer's comments, which have significantly enhanced the quality of the manuscript. The responses to each question and suggestion are presented below.

Reviewer 1

The paper described the changes in polyhenols, mineral composition and antioxidant activity occurring in lentil and quinoa samples submitted to different treatments following the digestion process which has been simulted in standard and elderly conditions.

It is interesting because the need to find value-added products for elderly is a real key problem of these and future years.

However, the manuscript need a revision according the following suggestions:

Abstract

“in Castellana 21 lentils from 5.05-10.5 mg gallic acid/g dry basis and in Pardina lentils from 6.6-7.5 mg gallic acid/g…”: from…to what?

The authors appreciate the feedback and have made the necessary changes for a better understanding of these lines.

“APCI” avoid to use abbreviation without before explaining its meaning

The authors have placed the full name in the abstract together with the abbreviation.

Results and discussion

Line 104 “highest” and not “higher”

The authors are grateful for the comment and have changed this word (line 107).

Lines 120-122: considering that only one lentil sample shows a significative difference between older adult digestive and standard conditions, what stated by the authors is excessive. In addition for white quinoa no significative difference is present. The sentence sgould be revised.

Effectively the paragraph was not correctly understood. The sentence has been revised and edited, and this idea has been attached to the previous paragraph in line 111.

Line 140: change “characterizarion”. It is no the right word

The authors have changed the word "characterization" to "The same chromatographic analysis" (line 136).

Line 155: “bioaccessible” and not “bio accessible”

The authors have rectified the orthographic error (line 156).

Lines 182-189: this part is not well linked to the below part od the section because the bioactivity tested is the antioxidant one

The authors agree with the comment; therefore, we have moved this paragraph to line 162 to connect that idea to the phenolic profile topic.

Lines 277-292: delete because it is a repetition

The authors appreciate the comment and have removed the duplicated text.

Add the chromatographic profile of a lentil o quinoa samples in the different treatment conditions in Supplementary material

The authors are grateful for the suggested improvement of the work, so we have added the chromatographic profile of the Pardina lentil digested samples with the different treatment conditions (line 137).

English language revisioni is suggested

English language revision has been made by an expert. The proper certificate is also attached in the cover letter.

Reviewer 2 Report

Comments and Suggestions for Authors

The results of this manuscript can provide some useful information. The workload of the manuscript is relatively comprehensive, but it is not sufficiently innovative and academically systematic, especially the Introduction and the Results analyses need to be strengthened in the writing. In addition, some deficiencies in the manuscript require major revisions. The specific comments are as follows:

1. Keywords need to be written in full, with initials capitalized.

2. The Introduction does not highlight the research topic as well as the focus. Much of the Introduction is conceptualized in terms of general background and does not focus on the main points of the study. It is recommended that the authors add a summary in the introduction as to why this research work was conducted and what the specific implications are. Also, the authors did not cite existing literature similar to this study as support for the background of the study.

3. Lines 41-66. The content is not well related and logical, resulting in confusion for the reader as to why this study was conducted. Failure to emphasize the content and focus of the study that is closely related to the research objectives of the manuscript.

4. Lines 102-124, 157-163. the use of fermentation and increased temperatures both have a positive effect on TPC in plants. In addition, the digestion effect of the aged digestion system is definitely reduced compared to the standard system. Most of these results are known to be predictable. So, the authors' aim was just to test the theory? What is the innovation of the manuscript?

5. Lines 376-380. Why two drying methods are used to dry fermented cereal seeds? The authors are requested to provide additional explanatory notes.

6. In the conclusion section, it is recommended to add the application of this study in real industrial production and a vision for future work.

Comments on the Quality of English Language

Extensive editing of English language are required.

Author Response

ANSWER TO REVIEWERS

The authors are very grateful for the reviewer's comments, which have significantly enhanced the quality of the manuscript. The responses to each question and suggestion are presented below.

Reviewer 2

The results of this manuscript can provide some useful information. The workload of the manuscript is relatively comprehensive, but it is not sufficiently innovative and academically systematic, especially the Introduction and the Results analyses need to be strengthened in the writing. In addition, some deficiencies in the manuscript require major revisions. The specific comments are as follows:

  1. Keywords need to be written in full, with initials capitalized.

The authors have made the suggested changes.

  1. The Introduction does not highlight the research topic as well as the focus. Much of the Introduction is conceptualized in terms of general background and does not focus on the main points of the study. It is recommended that the authors add a summary in the introduction as to why this research work was conducted and what the specific implications are. Also, the authors did not cite existing literature similar to this study as support for the background of the study.

The authors appreciate the valuable comments. To enhance the clarity of the introduction, its content has been restructured.

  1. Lines 41-66. The content is not well related and logical, resulting in confusion for the reader as to why this study was conducted. Failure to emphasize the content and focus of the study that is closely related to the research objectives of the manuscript.

The authors are grateful for comments and have made the necessary corrections.

  1. Lines 102-124, 157-163. the use of fermentation and increased temperatures both have a positive effect on TPC in plants. In addition, the digestion effect of the aged digestion system is definitely reduced compared to the standard system. Most of these results are known to be predictable. So, the authors' aim was just to test the theory? What is the innovation of the manuscript?

The authors appreciate the comment. However, we consider that the results of solid-state fermentation may vary depending on the specific substrate (plant material) and the microorganism used. The same effect of increased TPC has not always been seen in all legumes or cereals; there are also studies in which it has decreased. Therefore, it is essential to evaluate the results when lentils and quinoa are used together with the fungus Pleurotus ostreatus. To the authors' knowledge, no previous studies had been conducted on these plant materials in combination with P. ostreatus.

On the other hand, although it is true that there are expected results when digested under the altered conditions of the older adult's digestive system, there are few studies that evaluate the influence of this type of processing (fermentation and drying) after gastrointestinal digestion, especially in models of elderly digestion. For example, in the case of minerals, it was observed that there was no significant effect of the alterations that appear with ageing compared to the healthy adult. Furthermore, if the goal is to incorporate these fermented flours as novel ingredients in product development, it is imperative to understand their properties during digestion. The main idea of assessing fermented flours with the older adult digestion model was also to compare the substrates which one would be better in terms of increases or decreases in order to select the best fermented flour for this population group.

Below, you will find a list of works on fermentation that present variable results in Total Polyphenol Content (TPC). These fluctuations in TPC levels have been attributed to the specific substrates and fermentative microorganisms employed.

https://doi.org/10.1016/j.jcs.2018.05.008 (decrease)

https://doi.org/10.1016/j.foodchem.2007.05.017 (decrease)

https://doi.org/10.3390/molecules22122275 (increase depending on the substrate)

  1. Lines 376-380. Why two drying methods are used to dry fermented cereal seeds? The authors are requested to provide additional explanatory notes.

The authors have added additional explanatory text about the use of freeze-drying as a reference drying method on lines 360 and 361 for better understanding.

  1. In the conclusion section, it is recommended to add the application of this study in real industrial production and a vision for future work.

The authors appreciate the comment and agree with the reviewer's comment, thus for the recommendation to improve this work, the authors have added this suggestion at the end of the conclusions.

Round 2

Reviewer 1 Report

Comments and Suggestions for Authors

The paper has been improved following the Reviewer's comments and therefore now it is suitable for the publication

Reviewer 2 Report

Comments and Suggestions for Authors

The authors have addressed all the questions.